# Prevalence of Vertebral Fractures in CTPA’s in Adults Aged 75 and Older and Their Association with Subsequent Fractures and Mortality

**DOI:** 10.3390/geriatrics5030056

**Published:** 2020-09-21

**Authors:** Llewelyn Jones, Sukhdev Singh, Chris Edwards, Nimit Goyal, Inder Singh

**Affiliations:** 1Health Education and Improvement Wales (HEIW), Wales CF15 7QQ, UK; Llewelyn.Jones@wales.nhs.uk; 2Royal Gwent Hospital, Aneurin Bevan University Health Board, Newport NP20 4SZ, UK; Sukhdev.Singh@wales.nhs.uk (S.S.); Nimit.Goyal@wales.nhs.uk (N.G.); 3Department of Dermatology, St Wollas Hospital, Aneurin Bevan University Health Board, Newport N20 2UB, UK; Chris.Edwards3@wales.nhs.uk; 4Department of Geriatric Medicine, Ysbyty Ystrad Fawr, Aneurin Bevan University Health Board, Ystrad Mynach CF82 7EP, UK

**Keywords:** vertebral fracture, incidental, mortality, osteoporosis

## Abstract

Identifying vertebral fractures is prudent in the management of osteoporosis and the current literature suggests that less than one-third of incidental vertebral fractures are reported. The aim of this study is to determine the prevalence of reported and unreported vertebral fractures in computerized tomography pulmonary angiograms (CTPA) and their relevance to clinical outcomes. All acutely unwell patients aged 75 or older who underwent CTPAs were reviewed retrospectively. 179 CTPAs were reviewed to identify any unreported vertebral fractures. A total of 161 were included for further analysis. Of which, 14.3% (23/161) were reported to have a vertebral fracture, however, only 8.7% (14/161) of reports used the correct terminology of ‘fracture’. On subsequent review, an additional 19.3% (31/161) were noted to have vertebral fractures. Therefore, the overall prevalence of vertebral fractures was 33.5% (54/161). A total of 22.2% (12/54) of patients with a vertebral fracture on CTPA sustained a new fragility fracture during the follow-up period (4.5 years). In comparison, a significantly lower 10.3% (11/107) of patients without a vertebral fracture developed a subsequent fragility fracture during the same period (*p* = 0.04). Overall mortality during the follow-up period was significantly higher for patients with vertebral fractures (68.5%, 37/54) as compared to those without (45.8%, 49/107, *p* = 0.006). Vertebral fractures within the elderly population are underreported on CTPAs. The significance of detecting incidental vertebral fractures is clear given the increased rates of subsequent fractures and mortality. Radiologists and physicians alike must be made aware of the importance of identifying and treating incidental, vertebral fragility fractures.

## 1. Introduction

Osteoporosis is a skeletal disease of decreased bone mass and microarchitectural deterioration resulting in bone fragility and a susceptibility to fracture [1]. Fragility fractures, the clinical consequence of osteoporosis, are associated with significant morbidity and mortality with an estimated cost to the National Health Service (NHS) of £4.4 billion annually [2]. Identifying those at risk of fractures is the first step in initiating appropriate medical treatment, which is known to significantly reduce the risk of further fragility fractures [3]. The gold standard in identification of osteoporosis is bone densitometry using dual energy X-ray absorptiometry (DEXA). The Fracture Risk Assessment (FRAX) score, a validated cost-effective screening tool used to identify which patients require DEXA and/or treatment, is currently recommended within the UK by the National Institute for Health and Care Excellence (NICE) [4,5].

Identifying vertebral fractures is a key aspect in the diagnosis of osteoporosis as they occur more commonly than any other fragility fracture [6]. In addition to the FRAX score, including previous fractures as an input variable in their analysis, the treatment for osteoporosis can be commenced immediately following identification of a vertebral fracture if further investigation is not clinically appropriate. Unfortunately, due to the unspecific nature of their signs and symptoms, only one in four are clinically recognised [7]. Therefore, many vertebral fractures are first recognized incidentally.

Computerised tomography pulmonary angiograms (CTPAs) are frequently requested to confirm or exclude pulmonary embolisms. However, the thoracic vertebrae are clearly visualised in CTPAs and radiologists are expected to report any pathological bony finding. Thus, this commonly requested investigation could provide an invaluable tool in detecting clinically silent vertebral fractures. However, across a range of imaging modalities, less than one-third of incidental vertebral fractures are reported by radiologists [8]. Given up to 20% of patients over the age of 50 have a vertebral fracture, many opportunities to prevent further fractures are being missed [9]. The aim of this study is to determine the prevalence of reported and unreported vertebral fractures in CTPAs in adults aged 75 or older. An additional aim is to ascertain whether vertebral fractures on CTPA can identify patients at increased risk of subsequent fractures or mortality.

## 2. Materials and Methods

### 2.1. Study Design and Setting

This is a retrospective observational study. The study is based on the analysis of existing CTPAs for acutely unwell patients ages 75 or over, requested in Accident and Emergency or the medical assessment units of all three acute sites within Aneurin Bevan University Health board, Wales, UK. All CTPAs fitting this description between the 1st of January and 31st of December 2015 were reviewed and assessed for the presence of a vertebral facture. All patients were followed until June 2019 (4.5 years) for subsequent fractures and mortality.

Despite the reliance on imaging for accurate diagnosis, there is no internationally agreed radiological definition of a vertebral fracture, however several objective approaches have been described [10]. The standardised approach used for this study was the semiquantitative technique [11]. Firstly, all original CTPA reports were reviewed and reporting of vertebral fractures were noted, as well as the terminology used. All CTPA images were then reviewed by a fourth year medical student using sagittal reformats of the spine to identify vertebral fractures. All suspected fractures that were not initially reported were cross-checked by a consultant radiologist.

### 2.2. Data and Statistical Analysis

The electronic records for all patients, using the Welsh Clinical Portal (WCP), were reviewed for evidence of subsequent fracture, osteoporosis treatment history, date of death if applicable and evidence of a medical condition which may warrant patient exclusion. Sources of data included clinic letters, discharge summaries, General Practice records and investigation results.

Patients found to have a condition other than osteoporosis which could affect their bone integrity were excluded from further analysis. A total of 18 patients were excluded in total for the following reasons: cancers with bone metastases (10), myeloma (7) and a B-cell lymphoma causing a pathological fracture (1). All fractures, other than ankle, metatarsal or malleolar fractures were considered ‘fragility fractures’ and included in the analysis of subsequent fractures.

Two-tailed difference in proportion tests were performed to calculate the significance of the differences in incidence of subsequent fracture and mortality between those with and without vertebral fractures on CTPA. Kaplan–Meier survival curves were constructed with logrank and Breslow–Wilcoxon tests to analyse the impact of PE on mortality.

Ethical approval was not required as this was a service evaluation which does not constitute a research study. The service evaluation proposal was sent to the Aneurin Bevan University Health Board Research and Development Department where approval was granted.

## 3. Results

Data from 179 patient were gathered, 161 of which were included in the analysis. A total of14.3% (23/161) the CTPAs were reported to have a vertebral fracture, however only 14/161 (8.7%) used the correct terminology of ‘fracture’. The remaining 9/161 identified the fracture but used terminology such as: vertebral collapse, vertebral plana, vertebral compression or vertebral wedging. On subsequent review of each CTPA, an additional 31/161 (19.3%) vertebral fractures were found resulting in 54/161 (33.5%) fractures in total. This equated to 57.4% of all vertebral fractures going unreported (Table 1).

A significantly higher proportion of individuals with a vertebral fracture on CTPA sustained a subsequent fragility fracture (22.2%), whilst only 10.3% of those without a vertebral fracture on CTPA developed a subsequent fragility fracture (*p* = 0.04) (Table 2).

Sub-analysis for subsequent hip, vertebral and wrist fractures were done. Only 2.8% individuals sustained a new hip fracture without previous vertebral fracture on CTPA during follow up. However in comparison, 11.1% individuals with a previous vertebral fracture on CTPA sustained a subsequent hip fractures. This was significantly higher (*p* = 0.03). Overall, there is a higher chance of having a subsequent vertebral or a new wrist fracture if previously reported fracture on CTPA but this was not statistically significant (Table 2).

A significantly higher proportion (68.5%) of individuals with fracture on CTPA died as compared to those with no fracture identified on CTPA (45.8%, *p* < 0.01) with the 4.5 years follow-up. The Kaplan–Meier survival curve shown in Figure 1 demonstrates that incidental vertebral fractures found on CTPA are of significant prognostic importance. The cumulative survival of patients with vertebral fractures is less than those without at all time points. The mean survival of patients with a vertebral fracture on CTPA to be 873 days (95% CI: 702–1044) and without a vertebral fracture on CTPA to be 1148 days (95% CI: 1030–1265). Both the logrank and Breslow–Wilcoxon tests showed a significant difference between curves, *p* = 0.005 and *p* = 0.007, respectively and provides additional reason to identify patients with incidental vertebral fractures (Figure 1).

The data was then divided further to analyse the effects of vertebral fractures on mortality for the patients who were pulmonary embolism (PE) positive and negative on CTPA.

A total of 35 patients within the cohort were PE positive, of these 13 were positive for a vertebral fracture and 22 were negative. For this PE positive cohort, the mean survival for patients with a vertebral fracture was 696 days (95% CI: 327–1065) and the mean survival for patients without a vertebral fracture was 1283 days (95% CI: 1050–1516). Both logrank and Breslow–Wilcoxon tests showed a significant difference between curves, *p* = 0.017 and *p* = 0.011, respectively (Figure 2).

A total of 126 patients within the cohort were PE negative, of these 41 were positive for a vertebral fracture and 85 were negative. For this PE negative cohort, the mean survival for patients with a vertebral fracture was 917 days (95% CI: 732–1101) and the mean survival for patients without a vertebral fracture was 1112 days (95% CI: 978–1246). Neither logrank nor Breslow–Wilcoxon tests showed a significant difference between curves, *p* = 0.052 and *p* = 0.102, respectively (Figure 3).

Overall, vertebral fractures shown to reduce survival by a mean of 587 days, a statistically significant difference (*p* < 0.05). The larger cohort of 126 patients who were negative for PE on CTPA also showed that the presence of a vertebral fracture reduced survival by a mean of 195 days, however, this result did not achieve statistical significance.

Of the 54 patients with a vertebral fracture on CTPA, 51.9% (28/54) had a record of osteoporosis treatment on the Welsh Clinical Portal (alendronic acid (26), zolendronic acid (1) and denosumab (1)). Of these, 31.5% (17/54) had received osteoporosis treatment prior to the CTPA, it is not known if the physician prescribing this treatment was aware of the presence of the vertebral fracture from previous imaging or not. A total of 11.1% (6/54) of patients received osteoporosis treatment shortly after the CTPA, before any further fractures occurred. A further 9.3% (5/54) of patients received treatment for osteoporosis only after a subsequent fracture occurred. When looking only at the 14 patients who were initially reported to have a vertebral fracture on CTPA, using the correct terminology of ‘fracture’, 78.6% (11/14) had records of osteoporosis treatment: 8 prior to the CTPA, 1 after the CTPA and 2 only after a subsequent fracture occurred. If this is expanded to include all 23 patients who had a fracture reported on CTPA, with or without including the term ‘fracture’, 69.6% (16/23) of these patients had records of osteoporosis treatment: 10 prior to the CTPA, 3 following it and 3 only after a subsequent fracture occurred.

## 4. Discussion

Vertebral fractures commonly occur in patients with osteoporosis and are well known predictors of subsequent fractures, hospitalisation and mortality [12,13,14]. Their detection allows commencement of appropriate investigation and management of osteoporotic fractures. Our analysis shows the prevalence of vertebral fractures on CTPAs in patients aged 75 and over to be 33.5%. A large French study on women aged 75 or older showed a prevalence of 22.8% [15] and numerous other studies show the prevalence of vertebral fractures to be near 20% [6]. The discrepancy between our prevalence and that which is reported in the literature likely results from the differences in populations. Our cohort consisted of patients with a high average age of 82.1 years who were presenting acutely unwell.

It has previously been shown that incidental vertebral fractures are underreported. On CT scans of the thorax and/or abdomen, 9% to 16% of retrospectively confirmed vertebral fractures are initially reported [16,17,18,19]. Our data show that 42.6% of all vertebral fractures on CTPA were initially reported. Although our reporting rate is substantially higher than that seen in other studies, it is still surprisingly low for what is an easily recognisable finding. Indeed, when CTPAs are viewed in a midline sagittal section with bone windowing, recognising vertebral fractures requires little training. Similarly undiagnosed vertebral fractures have been reported in 9.2% (259) patients following retrospective review of routine 3216 chest radiographs. The prevalence of vertebral fracture was 2.4% in women aged 50–59 years and it increased to 8.9% in women aged 60–69 years and to 21.9% in women aged ≥70 years [20].

As such, we hypothesise that the low reporting rate by consultant radiologists is not due to an inability to detect the fractures, but rather not looking for vertebral pathologies given the clinical context of suspected pulmonary embolism. In many unreported cases it is possible the radiologists were aware of the vertebral fractures but decided not to report them, instead believing them to be unimportant, incidental findings. Therefore, to increase the reporting rate we believe radiologists must be made aware of the significance of incidental vertebral fractures.

Although our overall vertebral fracture reporting rate was 42.6%, not all of these included the correct terminology. Approaching figures quoted within the literature, only 25.9% of initial reports reported a vertebral ‘fracture’. The issue of variable and ambiguous terminology used to describe vertebral fractures is well recognised. To combat this, previous large studies have recommended the term ‘fracture’ be used in all cases [21]. We recommend that all vertebral fractures should be described as ‘vertebral fracture(s)’ by radiologists. Not only would this reduce misinterpretation by physicians but would also allow automated programmes to detect patients at risk of osteoporosis and automatically flag them for review by their GP or another bone health service.

Vertebral fractures are established predictors of subsequent fractures [12]. This study was also able to demonstrate this finding. We demonstrated a statistically significant relationship between the presence of a vertebral fracture on CTPA and an increased rate of subsequent fractures, which also held true when looking specifically at hip fractures. Importantly, there is high quality evidence that bisphosphate therapy is effective at reducing future fracture risk in those with osteoporosis on DEXA or a history of vertebral fracture(s) [3].

This study highlights the importance of detecting incidental vertebral fractures as it provides the opportunity to initiate appropriate care plan for management of osteoporotic fractures. Diagnosis prevalent fractures does lead to higher pressure on current health care and there is a need to explore integrated working and implementing quality initiatives [22]. Prevalence of fragility fracture in older women above 65 has been reported as 20% and 47% are not prescribed any treatment for osteoporosis [23]. There is a wide range of treatment options available including standard first line antiresorptive treatment, monoclonal antibody (Denosumab) slows down the natural rate bone break down, biological anabolic medicines that promote bone formation (Teriparatide) [4]. In addition, there are surgical options including implant fixation or percutaneous vertebroplasty [24,25]. Multidisciplinary team input including management of falls risk, muscle and balance strengthening and polypharmacy review remains an essential component to provide holistic person centred approach.

This study also found the presence of vertebral fractures on CTPA to be significantly associated with shortened survival. When analysed further, whilst the trend persisted, the statistical significance of this was lost if the patient was negative for a PE. For patients who were PE positive the same trend remained statistically significant. As such, identifying patients with incidental vertebral fractures could have benefits outside of simply identifying patients requiring osteoporosis treatment. For example, by identifying patients at risk of frailty and/or requiring a comprehensive geriatric assessment.

This study has also shown that even when vertebral fractures are identified and reported by radiologists, 43.5% of these patients either have no record of prior or subsequent osteoporosis treatment or only receive it following a subsequent fracture. Further analysis to determine the beneficial effects of treating vertebral fracture positive patients, with regards to future fracture risk and/or mortality, was not undertaken due to the limited cohort size. Nevertheless, it is clear that even when fractures are reported on CTPAs, opportunities to apply the osteoporosis treatment guidelines are being missed. A larger study is required to analyse the beneficial effects osteoporosis treatment would have at this stage.

Limitations to this study included data collection using the WCP. Although the WCP allowed viewing of all clinical letters and discharge summaries since the late 1990s we did not have access to GP records unless a GP had made a referral within this time, which was not always the cases. Additionally, we may have missed subsequent fractures or even date of death if they occurred outside of Wales. The size of the study was a further limitation, which became particularly apparent when analysing the data for the different types of subsequent fractures and when dividing the cohort into those who were PE positive and negative. CTPA’s were initially reviewed by a fourth year medical student. However, any discrepancy between the initial radiologist’s report and the medical student’s findings were cross-checked by a consultant radiologist. We were not able to record patient characteristics or past medical history as part of this retrospective study, therefore, recommend these to be included for subsequent studies. Although this study showed that prevalent vertebral fractures are associated with subsequent fractures and mortality, these findings need to be confirmed with further studies, which adjust for other variables and confounding factors.

## 5. Conclusions

Vertebral fractures on CTPA’s in patients aged 75 or older are underreported. When reported, there is a variety of terminology used. The significance of this finding is in the ability of these incidental vertebral fractures to identify those at risk of subsequent fractures and death. Addressing this issue by informing radiology departments of the importance of diagnosing incidental vertebral fractures on CTPAs is required. When incidental vertebral fractures are reported, an appropriate osteoporosis care plan must be arranged with the patient’s primary or secondary care physician.

## Figures and Tables

**Figure 1 geriatrics-05-00056-f001:**
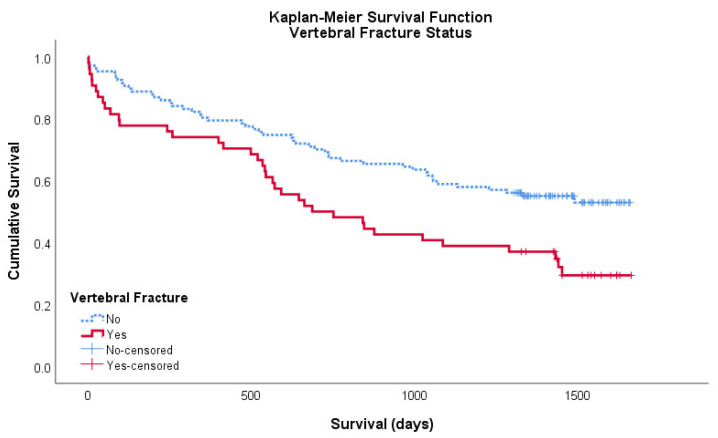
Kaplan–Meier survival curve showing the cumulative survival of patients with (red/solid) and without (blue/broken) a vertebral fracture on CTPA. Censored data is shown as ‘+’ in its respective colour.

**Figure 2 geriatrics-05-00056-f002:**
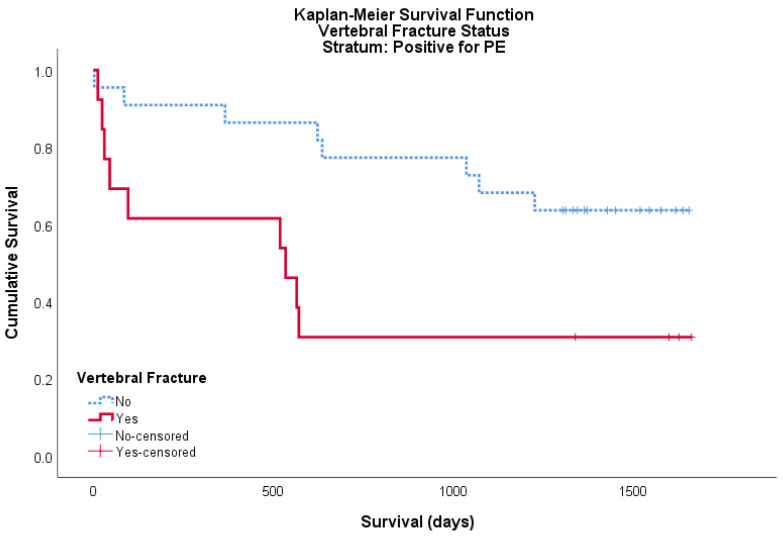
Kaplan–Meier survival curve showing the cumulative survival of patients with (red/solid) and without (blue/broken) a vertebral fracture on CTPA for patients positive for pulmonary embolism (PE). Censored data is shown as ‘+’ in its respective colour.

**Figure 3 geriatrics-05-00056-f003:**
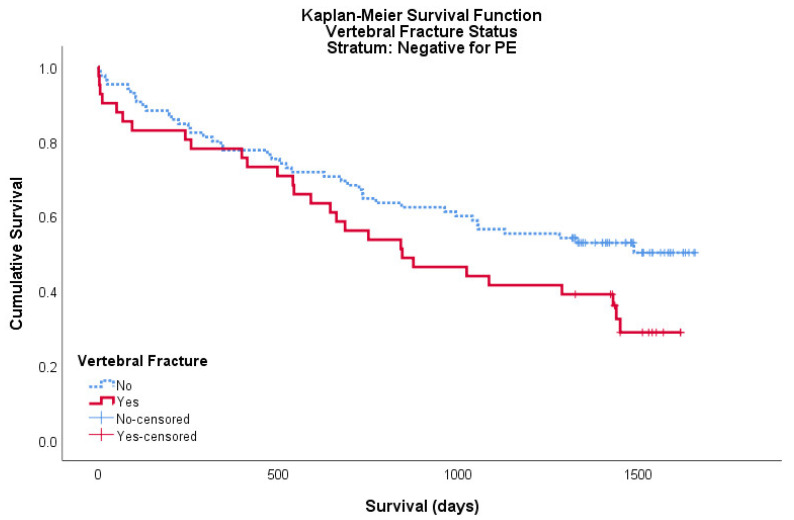
Kaplan–Meier survival curve showing the cumulative survival of patients with (red/solid) and without (blue/broken) a vertebral fracture on CTPA for patients negative for PE. Censored data is shown as ‘+’ in its respective colour.

**Table 1 geriatrics-05-00056-t001:** Computerized tomography pulmonary angiograms (CTPA) reports and fracture incidence.

CTPA Reports	Total	% of Cohort	% of Fractures
Cohort size	161	/	/
Fracture present	54	33.5%	/
Unreported fracture	31	19.3%	57.4%
Reported vertebral fracture: All terminology	23	14.3%	42.6%
Reported vertebral fracture: Term ‘fracture’ used	14	8.7%	25.9%

**Table 2 geriatrics-05-00056-t002:** Subsequent fractures and mortality following CTPA.

	Vertebral Fracture	No Vertebral Fracture	*p*
Subsequent fracture: Hip	6/54 (11.1%)	3/107 (2.8%)	0.0303
Subsequent fracture: Vertebral	3/54 (5.6%)	3/107 (2.8%)	0.3843
Subsequent fracture: Wrist	2/54 (3.7%)	3/107 (2.8%)	0.7566
Subsequent fracture: All	12/54 (22.2%)	11/107 (10.3%)	0.0414
Mortality by 26 June 2019	37/54 (68.5%)	49/107 (45.8%)	0.0063

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
