# Peer review of "Prevalence of Vertebral Fractures in CTPA’s in Adults Aged 75 and Older and Their Association with Subsequent Fractures and Mortality"

_geriatrics, 2020, doi:10.3390/geriatrics5030056_

Round 1
Reviewer 1 Report
The topic is very interesting as osteoporotic fractures represent a very common disease. Results are good. References must be improved, as they are not so recent. Please, carefully look at these 3 recent references also in setting of spine biomechanics
- The Y-shaped trabecular bone structure in the odontoid process of the axis: a CT scan study in 54 healthy subjects and biomechanical considerations [published online ahead of print, 2019 Feb 1]. J Neurosurg Spine. 2019;1-8. doi:10.3171/2018.9.SPINE18396
- Biomechanics of Implant Fixation in Osteoporotic Bone [published online ahead of print, 2020 Jul 30]. Curr Osteoporos Rep. 2020;10.1007/s11914-020-00614-2. doi:10.1007/s11914-020-00614-2
- Regional and experiential differences in surgeon preference for the treatment of cervical facet injuries: a case study survey with the AO Spine Cervical Classification Validation Group [published online ahead of print, 2020 Jul 22]. Eur Spine J. 2020;10.1007/s00586-020-06535-z. doi:10.1007/s00586-020-06535-z
In the conclusion: "an appropriate osteoporosis care plan must be arranged with the patient’s primary or secondary care physician", can you explain better about this medical plan in the discussion?
Author Response
Dear Reviewer,
Many thanks for taking time to give us feedback. Please find our responses as below
- We have added a new reference that discuss a similar theme on prevalent fracture based on the review of chest x-rays.
- As you suggested, we have discussed osteoporosis care plan in the discussion and we have used one of the suggested reference to expand on osteoporosis management with regard to implant fixation in osteoporotic bone. A new paragraph has been added.
This study highlights the importance of detecting incidental vertebral fractures as it provides the opportunity to initiate appropriate care plan for management of osteoporotic fractures. Diagnosis prevalent fractures does lead to higher pressure on current health care and there is a need to explore integrated working and implementing quality initiatives [22]. Prevalence of fragility fracture in older women above 65 has been reported as 20% and 47% are not prescribed any treatment for osteoporosis [23]. There is a wide range of treatment options available including standard first line antiresorptive treatment, monoclonal antibody (Denosumab) slows down the natural rate bone break down, biological anabolic medicines that promote bone formation (Teriparatide) [4]. In addition, there are surgical options including implant fixation or percutaneous vertebroplasty [24, 25]. Multidisciplinary team input including management of falls risk, muscle and balance strengthening and polypharmacy review remains an essential component to provide holistic person centred approach.
We thank you again for you time, please do not hesitate to contact us if you have any further queries.
Reviewer 2 Report
The aim of
58 this study is to determine the prevalence of reported and unreported vertebral fractures in CTPA’s
59 in adults aged 75 or older. An additional aim is to ascertain whether vertebral fractures on CTPA can
60 identify patients at increased risk of subsequent fractures or mortality.
minor revision:
1: although the impact of PE on mortality is important, why did you select PE among many factors related with death?
major revision:
1:fragility fracture occur the following order: wrist, proximal humerus, vertebral, hip.
As table 2, you check subsequent fracture following vertebral fracture.
Did you check fracture prior to vertebral fracture?
2: inclusion criteria is the existing CTPA’s.
I think patient with CTPA is at the risk of cardiovascular disease.
Please show the patient characteristic on past medical history.
3: how do you confirm the effect of vertebral fracture without adjusting covaribles?
Author Response
Dear Reviewer,
Many thanks for taking time to give us feedback. Please find our responses as below
- We agree with you that there could be other factors that could be predictors of mortality but it was not possible to capture that data in this retrospective study and to minimise the bias we have only measured impact of PE on mortality. We agree with your views and have mentioned this as a limitation whilst interpreting impact of PE on mortality.
- We did not check fractures (wrist or humerus) prior to vertebral fractures as number was so small and we have mentioned small power of this study as limitation. We have added more clarity in our discussion with regard to your comments.
- We agree with our view that patients who had positive CTPA or require CTPA does have other associated cardiovascular risk factors. We were not able to record patient characteristics or past medical history as part of this retrospective study and have accepted this as a limitation of the study. We have included your comment as a recommendation for subsequent studies.
- We can only show association but cannot confirm the effect of vertebral fracture without adjusting co-variables, which has not been recorded and we have included this both as a limitation of this study and a recommendation.
We thank you again for you time, please do not hesitate to contact us if you have any further queries.